# Proportion of paediatric admissions with any stage of noma at the Anka General Hospital, northwest Nigeria

**Elise Farley**[1]*, **Miriam Njoki Karinja**[2], **Abdulhakeem Mohammed Lawal**[2], **Michael Olaleye**[1], **Sadiya Muhammad**[2], **Maryam Umar**[2], **Fatima Khalid Gaya**[2], **Shirley Chioma Mbaeri**[2], **Mark Sherlock**[3], **Deogracia Wa Kabila**[1], **Miriam Peters**[2], **Joseph Samuel**[1], **Guy Maloba**[2], **Rabi Usman**[4], **Saskia van der Kam**[3], **Koert Ritmeijer**[3], **Cono Ariti**[5], **Mohana Amirtharajah**[3], **Annick Lenglet**[3], **Grégoire Falq**[3]

**1** Médecins Sans Frontières, Noma Children's Hospital, Sokoto, Nigeria, **2** Médecins Sans Frontières, Nigeria Mission, Zamfara and Abuja, Nigeria, **3** Médecins Sans Frontières, Amsterdam, Netherlands, **4** Zamfara Ministry of Health, Zamfara, Nigeria, **5** London School of Hygiene and Tropical Medicine, London, United Kingdom

* elisefarley@gmail.com

**Data Availability Statement:** MSF has a managed access system for data sharing that respects MSF's legal and ethical obligations to its patients to collect, manage and protect their data

## Abstract

### Introduction

Noma is a rapidly spreading infection of the oral cavity which mainly affects young children. Without early treatment, it can have a high mortality rate. Simple gingivitis is a warning sign for noma, and acute necrotizing gingivitis is the first stage of noma. The epidemiology of noma is not well understood. We aimed to understand the prevalence of all stages of noma in hospitalised children.

### Methods

We conducted a prospective observational study from 1st June to 24th October 2021, enrolling patients aged 0 to 12 years who were admitted to the Anka General Hospital, Zamfara, northwest Nigeria. Consenting parents/ guardians of participants were interviewed at admission. Participants had anthropometric and oral examinations at admission and discharge.

### Findings

Of the 2346 patients, 58 (2.5%) were diagnosed with simple gingivitis and six (n = 0.3%) with acute necrotizing gingivitis upon admission. Of those admitted to the Inpatient Therapeutic Feeding Centre (ITFC), 3.4% (n = 37, CI 2.5–4.7%) were diagnosed with simple gingivitis upon admission compared to 1.7% of those not admitted to the ITFC (n = 21, CI 1.1–2.6%) (p = 0.008). Risk factors identified for having simple gingivitis included being aged over two years (2 to 6 yrs old, odds ratio (OR) 3.4, CI 1.77–6.5; 7 to 12 yrs OR 5.0, CI 1.7–14.6; p = <0.001), being admitted to the ITFC (OR 2.1; 1.22–3.62) and having oral health issues in the three months prior to the assessment (OR 18.75; CI 10.65, 33.01). All (n = 4/4) those aged six months to five years acute necrotizing gingivitis had chronic malnutrition.

responsibility. Ethical risks include but are not limited to the nature of MSF operations and target populations being such that data collected often involves highly sensitive data. The dataset supporting the conclusions of this article is available on request in accordance with MSF's data sharing policy. Requests for access to data should be made to data.sharing@msf.org.

**Funding:** The author(s) received no specific funding for this work.

**Competing interests:** The authors declare no conflict of interest.

## Conclusion

Our study showed a small proportion of children admitted to the Anka General Hospital had simple or acute necrotizing gingivitis. Hospital admission with malnutrition was a risk factor for both simple and acute necrotizing gingivitis. The lack of access to and uptake of oral health care indicates a strong need for oral examinations to be included in routine health services. This provision could improve the oral status of the population and decrease the chance of patients developing noma.

## Author summary

Noma is a rapidly spreading infection of the oral cavity which mainly affects young children. Without early treatment, it can have a high mortality rate. Simple gingivitis is a warning sign for noma, and acute necrotizing gingivitis is the first stage of noma. We aimed to gather evidence on the epidemiology of noma and its association with malnutrition by conducting a prospective observational study enrolling 2346 patients aged 0 to 12 years who were admitted to the Anka General Hospital, Zamfara, northwest Nigeria. Consenting parents/ guardians of participants were interviewed at admission. Patients had anthropometric and oral exams at admission and discharge. Our study showed a small proportion of those admitted to the Anka General Hospital had simple or acute necrotizing gingivitis. Those admitted to the ITFC were more likely to have simple gingivitis. All of those diagnosed with acute necrotizing gingivitis had chronic malnutrition and most had acute malnutrition. The lack of access to and uptake of oral health care indicates a strong need for oral examinations to be included in routine health services. This provision could improve the oral status of the population and decrease the chance of patients developing noma.

## Introduction

Noma is an infection of the oral cavity which can cause the disintegration of the cheek, nose and/or eye in a few weeks [1]. If untreated in the early reversible stages, noma has a reported 90% mortality rate [2] and mainly affects children aged between two and six years [3]. Deaths in noma patients are primarily due to starvation, aspiration pneumonia, respiratory insufficiency or sepsis [4,5]. With timely antibiotic treatment, wound debridement and nutritional support, morbidity and mortality from noma greatly decrease [1]. For those who survive, noma often leads to severe facial disfigurement, functional issues and stigmatization [1,2,6,7]. The WHO classifies noma into stages: Warning Sign: simple gingivitis, Stage 1: acute necrotizing gingivitis, Stage 2: oedema, Stage 3: gangrene, Stage 4: scarring and Stage 5: sequelae [2].

There is limited research on noma and as a result, many gaps in knowledge exist around the disease [1]. The aetiology of noma is unknown but thought to be multifactorial [1]. Risk factors for noma include poor oral hygiene, limited access to quality health care including routine childhood vaccinations, low socioeconomic status and immunosuppression resulting from comorbidities such as malnutrition, measles and HIV [1,8–17]. Malnutrition is frequently listed as a risk factor for noma, however, evidence supporting this theory is largely based on case reports and case series [12,18–28] and a handful of primary studies [8–11,29–31]. The epidemiology of the disease is also not well understood. The WHO estimates that 140,

000 children contract noma each year globally [32]. In 2003, a northwest Nigerian study estimated the noma incidence was 6.4 per 1000 children from 1996 to 2001 [33]. A recent study estimated that the period prevalence of noma from 2010–2018 was 1.6 per 100,000 population at risk in northern Nigeria [34]. A 2019 prevalence study in Sokoto and Kebbi states, Nigeria, identified that in children aged 0 to 15 years (n = 7122), simple gingivitis was diagnosed in 3.1% (n = 181; 95% confidence interval (CI) 2.6–3.8), acute necrotizing gingivitis in 0.1% (n = 10; CI 0.1–0.3), and oedema in 0.05% (n = 3; CI 0.02–0.2), no late stage noma cases were identified [35].

We conducted a prospective observational study assessing the proportion of paediatric admissions at the Anka General Hospital, Zamfara, northwest Nigeria with any stage of noma. Depending on their nutritional status at admission, paediatric patients are admitted to two wards: either the general paediatric ward or the Inpatient Therapeutic Feeding Centre (ITFC). The ITFC is for the treatment of children with severe acute malnutrition combined with medical complications.

## Methods

### Ethics statement

The Médecins Sans Frontières Ethics Review Board (ERB) (2017), Nigerian Federal Ministry of Health ERB (NHREC/01/01/2007-29/04/2021), Zamfara Ministry of Health ERB (ZSHREC01112020) and the Usman Danfodiyo University Teaching Hospital Health Research and Ethics Committee in Nigeria (NHREC/30/012/2019) approved the study protocol.

Formal written consent was obtained from parents/ guardians of all participating patients and patient participants aged between seven and 12 years were asked to provide formal written assent. For participants or parents/ guardians who were illiterate, we requested them to provide an initial and a thumb print as a representation of consent/ assent, with additional participation from an independent literate witness who checked that the parent/ guardian/ participant understood the contents of the consent form. This witness co-signed the consent form. The information sheet and the informed consent and assent forms were written in English and Hausa.

### Study design and data collection

From 1st June to 24th October 2021, we conducted a prospective observational study utilising face-to-face interviews with parents/ guardians of paediatric patients (aged under 12 years) admitted to the Anka General Hospital, northwest Zamfara, Nigeria and physical examinations of the patients themselves. All parents/ guardians of patients admitted between 09h00 and 16h00 Monday to Sunday were approached on the day of admission by the study nurse. Consenting individuals were enrolled (discussed in the ethics statement above). A tablet-based RedCap data collection form was used to collect the study specific data on admission (sociodemographic characteristics, infant and current feeding practices, oral health practices, health care access, oral examination, mid upper arm circumference, weight, and height) and at discharge (oral exam and access to routinely collected medical records). Patients were treated according to Ministry of Health and Médecins Sans Frontières paediatric protocols for the duration of their stay in the hospital.

### Sample size

We aimed to have a sample size large enough to estimate the proportion of paediatric admissions with any stage of noma admitted to the ITFC or other wards. This calculation was based on the following assumptions:

- 6%- Inpatient department proportion of patients diagnosed with any stage of noma during an operational research survey conducted by MSF (as this had a lower proportion of any stage of noma diagnosed compared to the ITFC [36])

- 95% confidence interval

- 1.5% margin of error (i.e., can estimate a proportion between 4.5 and 7.5% assuming a true proportion of 6%)

The sample size calculation showed that it was necessary to include 963 patients in the ITFC and 963 patients in the other wards.

### Analysis

We performed a descriptive analysis of the data collected. Categorical variables are shown as frequencies and percentages. Continuous variables were summarised using means and standard deviations (SD) or medians and interquartile ranges (IQR) depending on normality. Missing data numbers are recorded in each table.

Age was categorised into 0–1 years, 2–6 years and 7–12 years. This age categorisation was selected as children aged between two and six years have previously been reported to be at higher risk of noma [3].

Malnutrition was categorised based on the "Consensus Statement of the Academy of Nutrition and Dietetics/American Society for Parenteral and Enteral Nutrition" [37] and the 2007 WHO Growth Reference Data, as:

1. Those aged six months to five years who had chronic malnutrition/ were stunted (height-for-age z-score $< -2$ SD);

2. Those aged six months to five years who had acute malnutrition (weight-for-height z-score $< -2$ SD and/ or arm-circumference-for age z-score $< -2$SD), and;

3. Those aged six to 12 years who had moderate/ severe acute malnutrition ($< -2$ BMI-for-age z-score).

Wealth scores were calculated by assigning a value of one to each of the following items owned by the family: cell phone, radio, motorbike, tractor, bicycle, car (these items were chosen based on consultation with local researchers, knowledgeable about the context). The minimum wealth score was zero and the maximum was six, all items weighed equally in the scoring.

We calculated the number of admissions who had any stage of noma upon admission, and the outcome of those patients (noma stage at discharge) for all those who had oral examinations on both admission and discharge.

We compared the proportion of admissions by malnutrition status using a Pearson's chi-square test.

We assessed risk factors for simple gingivitis using univariate logistic regression (patient age, wealth score, ITFC admission status, malnutrition status, if the child was sick during the past three months, if the child had oral health issues in the past three months, if the respondent was the primary caretaker, if the child had been given colostrum at birth, if the child eats pap, if the child was vaccinated, measles, and malaria diagnoses). These variables were selected as they are reported risk factors for noma [1,8,10–28]. We conducted an exploratory analysis of risk factors for acute necrotizing gingivitis; proportions are reported.

All data analysis was conducted with Stata 17 (StataCorp LP, College Station, TX, USA).

## Results

### Sociodemographic characteristics

Of those parents/ guardians approached (n = 2356), six did not provide consent and four did not provide assent to be part of the study; thus 2346 participants were enrolled and included in the analysis, 2340 of these had a discharge assessment. Parents/ guardians were mainly aged between 18 and 25 years (n = 1036, 44.3%) and 26 to 35 years (n = 946, 40.5%), female (n = 2318, 99.2%) and were the mother of the child (n = 2211, 94.6%). The main source of income of most families was agriculture (n = 1285, 55.0%). Most (n = 1696, 72.3%) families owned zero to two items on the wealth score list. Half the child participants were female (n = 1181, 50.6%), most were aged between zero and one years (n = 1097, 47.1%) or two to six years (n = 1135, 48.8%) (Table 1).

### Health care access and primary diagnoses

Parents/ guardians reported several challenges accessing care including difficulties with transport (n = 454, 19.4%) and the hospital being far from their home (n = 432, 18.4%). Half of the respondents (n = 1176, 50.4%) reported that the child had been sick in the prior three months (not including the illness that made them seek care on the day of enrolment). Of these, almost all (n = 1069, 90.9%) sought care for that illness, many from a street pharmacist (n = 424, 39.7%) (Table 2).

Of those respondents who had teeth (n = 1907, 81.7%), almost all (n = 1652, 86.6%) cleaned their teeth, mainly using water (n = 1399, 84.7%). The main reasons for children cleaning their teeth were to get clean, bright teeth (n = 537, 32.5%) and to get rid of foul breath (n = 944, 57.1%). The main reasons for children not cleaning their teeth (n = 251), were not knowing the benefits of cleaning (n = 18, 7.2%) and forgetting to clean their teeth (n = 19, 7.6%). Most children (n = 2069, 88.5%) had not had any oral issues in the three months prior to study enrolment. The most common oral health issues reported (n = 265, 11.4%) were a sore in the mouth (n = 149, 56.2%), bleeding gums when touched (n = 88, 33.2%) and sore gums (n = 34, 12.8%). Almost all the respondents (n = 2274, 97.4) had not had an oral health check in the past year (Table 3).

The most common primary diagnosis of patients admitted to the hospital were malaria (n = 1086, 46.3%) and severe acute malnutrition (n = 977, 41.6%). The median number of days patients spent in the hospital was three (IQR 2, 5) (Table 4).

### Simple and acute necrotizing gingivitis diagnoses

Simple gingivitis was diagnosed in 58 patients (2.5%) and acute necrotizing gingivitis was diagnosed in six patients (0.3%) at admission. At discharge simple gingivitis was diagnosed in nine patients (0.5%) and acute necrotizing gingivitis was diagnosed in one patient (0.1%) (Table 5).

Of the 34 participants who were diagnosed with simple gingivitis on admission and who had an oral exam at discharge: 28 (82.4%) had resolved at discharge; five (14.7%) still had simple gingivitis at discharge; and one (2.9%) had acute necrotizing gingivitis upon discharge. One of the two patients who had acute necrotizing gingivitis at admission and had an oral exam at discharge, had no stage of noma at discharge, the other had simple gingivitis at discharge.

### Malnutrition

Those who had chronic malnutrition and were aged between six months and five years were more likely to be diagnosed with simple or acute necrotizing gingivitis in comparison to those who did not have chronic malnutrition (63.5% vs 49.7%; p = <0.05) (Table 6).

**Table 1. Sociodemographic characteristics, noma study Anka General Hospital 2021.**

| Caretaker age | N = 2346 | % |
|---|---|---|
| 18–25 years | 1036 | 44.3% |
| 26–35 years | 946 | 40.5% |
| 36–45 years | 283 | 12.1% |
| > 46 years | 71 | 3.1% |
| Missing | 10 | |
| Caretaker sex | | |
| Female | 2318 | 99.2% |
| Male | 19 | 0.8% |
| Missing | 9 | |
| Caretaker education | | |
| None | 732 | 31.3% |
| Arabic studies | 1499 | 64.1% |
| Primary school | 54 | 2.3% |
| Secondary school | 50 | 2.1% |
| University | 2 | 0.1% |
| Missing | 9 | |
| Income | | |
| Agriculture | 1285 | 55.0% |
| Trader | 745 | 31.9% |
| Mining | 88 | 3.8% |
| Daily labourer | 53 | 2.3% |
| Civil servant | 43 | 1.8% |
| Livestock | 37 | 1.6% |
| Other | 85 | 3.6% |
| Missing | 10 | |
| Relationship to child | | |
| Mother | 2211 | 94.6% |
| Other | 125 | 5.4% |
| Missing | 10 | |
| Wealth score (own: cell phone, radio, motorbike, tractor, bicycle, car) | | |
| 0–2 items | 1696 | 72.3% |
| 3–6 items | 650 | 27.7% |
| Child gender | | |
| Female | 1181 | 50.6% |
| Male | 1154 | 49.4% |
| Missing | 11 | |
| Child age (years) | | |
| 0–1 | 1097 | 47.1% |
| 2–6 | 1135 | 48.8% |
| 7–12 | 95 | 4.1% |
| Missing | 19 | |

## Risk factors

Being aged over two years (2-6yr old odds ratio (OR) 3.4, CI 1.77, 6.5; 7 to 12 yr OR 5.0, CI 1.7, 14.6; p = <0.001), being admitted to the ITFC (OR 2.1; CI 1.22, 3.62) and having oral health issues in the three months prior to the assessment (OR 18.75; CI 10.65, 33.01) were associated with having simple gingivitis. Those whose primary diagnosis was malaria, were less likely to

**Table 2. Health care access noma study Anka General Hospital 2021.**

| Challenges when accessing care | N = 2346 | % |
|---|---|---|
| Transport difficult | 454 | 19.4% |
| Far to get to hospital | 432 | 18.4% |
| Difficult to leave other children | 22 | 0.9% |
| Takes a long time to get to hospital | 52 | 2.2% |
| Child very sick, difficult to travel | 6 | 0.3% |
| Difficult to get permission from family to travel | 6 | 0.3% |
| Cost of care too expensive | 13 | 0.6% |
| No challenges | 1505 | 64.2% |
| Has your child been sick in the past 3 month? (other than the sickness that made you seek care today) | | |
| No | 1157 | 49.6% |
| Yes | 1176 | 50.4% |
| Missing | 13 | |
| If yes, did you seek care due to that sickness? (N = 1176) | | |
| No | 104 | 8.8% |
| Yes | 1069 | 90.9% |
| Where did you seek care? (N = 1069) | | |
| Hospital | 297 | 27.8% |
| Clinic | 327 | 30.6% |
| Traditional healer | 62 | 5.8% |
| Street chemist, pharmacist | 424 | 39.7% |

**Table 3. Oral hygiene and health care noma study Anka General Hospital 2021.**

| Does your child have any teeth? | N = 2346 | % |
|---|---|---|
| No | 428 | 18.3% |
| Yes | 1907 | 81.7% |
| Missing | 11 | |
| Does your child clean their teeth? (N = 1907) | | |
| Yes | 1652 | 86.6% |
| No | 251 | 13.2% |
| Don't know | 4 | 0.2% |
| Reason for not cleaning (N = 251) | | |
| No time for brushing | 12 | 4.8% |
| No money to buy toothbrush and toothpaste | 1 | 0.4% |
| Nobody brushes their teeth in my family | 2 | 0.8% |
| Don't know of any benefits from brushing | 18 | 7.2% |
| Gums are bleeding when brushing | 1 | 0.4% |
| Useless, good teeth are hereditary | 6 | 2.4% |
| Always forget to brush my teeth | 19 | 7.6% |
| Reason for cleaning (N = 1652) | | |
| Clean, bright teeth | 537 | 32.5% |
| Prevention of caries | 279 | 16.9% |
| Prevention of bleeding gums | 15 | 0.9% |
| Prevention of oral ulcer | 107 | 6.5% |
| To get rid of foul breath | 944 | 57.1% |
| Cleaning utensil (N = 1652) | | |
| Toothbrush | 171 | 10.4% |

(*Continued*)

**Table 3.** (Continued)

| Does your child have any teeth? | N = 2346 | % |
|---|---:|---:|
| Stick | 2 | 0.1% |
| Cloth | 29 | 1.8% |
| Ash | 39 | 2.4% |
| Salt and water | 110 | 6.7% |
| Water | 1399 | 84.7% |
| Does your child use toothpaste? (N = 1652) | | |
| No | 1418 | 85.8% |
| Yes | 216 | 13.1% |
| No times teeth clean/day (Median, IQR) (N = 1652) | 1 | 1,2 |
| How long does your child clean their teeth for? (N = 1652) | | |
| Less than 3 minutes | 1580 | 95.6% |
| 3 minutes or more | 61 | 3.7% |
| Child oral health issues in the past three months? | N = 2346 | % |
| No | 2069 | 88.6% |
| Yes | 265 | 11.4% |
| Missing | 12 | |
| Oral issues past 3 months (N = 265) | | |
| Sore gums | 34 | 12.8% |
| Bleeding gums when touched | 88 | 33.2% |
| Spontaneously bleeding gums | 12 | 4.5% |
| Sore in mouth | 149 | 56.2% |
| Pus in mouth | 4 | 1.5% |
| Sore jaw | 8 | 3.0% |
| Tooth pain | 10 | 3.8% |
| Has your child visited an oral health care provider for a check-up in the past year | | |
| No | 2274 | 97.4% |
| Yes | 60 | 2.6% |
| Missing | 12 | |
| Oral health provider (N = 60) | | |
| Hospital | 10 | 16.7% |
| Clinic | 26 | 43.3% |
| Traditional healer | 14 | 23.3% |
| Street chemist, pharmacist | 13 | 21.7% |

**Table 4. Diagnosis noma study Anka General Hospital 2021.**

| Primary diagnosis | N = 2346 | % |
|---|---:|---:|
| Malaria | 1086 | 46.3% |
| Severe acute malnutrition | 977 | 41.6% |
| Measles | 114 | 4.9% |
| Other | 56 | 2.4% |
| Lower respiratory tract infection | 50 | 2.1% |
| Acute watery diarrhoea | 36 | 1.5% |
| Other | 15 | 0.5% |
| Length of stay in hospital (days) (Median, IQR) | 3 | 2, 5 |
| Length of stay in hospital (days) (range) | 0 | 97 |

**Table 5. Noma diagnosis upon admission and discharge, noma study Anka General Hospital 2021.**

| Admission Noma Stage | N = 2346 | % |
|---|---|---|
| None | 2256 | 97.2% |
| Stage 0: Simple gingivitis | 58 | 2.5% |
| Stage 1: Acute necrotizing gingivitis | 6 | 0.3% |
| Missing | 26 | |
| Discharge Noma Stage | | |
| None | 1667 | 99.4% |
| Stage 0: Simple gingivitis | 9 | 0.5% |
| Stage 1: Acute necrotizing gingivitis | 1 | 0.1% |
| Missing | 669 | |

have simple gingivitis upon admission compared to those admitted for other reasons (OR 0.29; CI 0.15, 0.55; p = <0.001).

Of those diagnosed with acute necrotizing gingivitis, almost all (n = 5/6) had been sick in the 3 months prior to admission, and most (n = 4/6) were aged 2–6 years, had oral health issues in the prior three months, were in the lower wealth score (indicating lower socioeconomic status) and ate pap. and All those aged six months to five years (n = 4/4) had chronic malnutrition (Table 7).

## Discussion

A small proportion of those enrolled in this study had simple gingivitis (n = 58, 2.5%) or acute necrotizing gingivitis (n = 6, 0.3%) at admission, and no later stages of noma were identified in the cohort. Oral health practices were not optimal and access to oral health care was limited and underutilised. Those with chronic malnutrition and those in the ITFC were more likely to have simple or acute necrotizing gingivitis. Known risk factors for noma (malnutrition, prior oral health issues and being aged between two and six years) were associated with simple gingivitis in our cohort.

Our findings were similar to a community-based noma study in northwest Nigeria in 2020 where simple gingivitis was identified in 3.1% (n = 181; 95% CI 2.6 to 3.8), acute necrotising gingivitis in 0.1% (n = 10; CI 0.1 to 0.3) and oedema in 0.05% (n = 3; CI 0.02 to 0.2) [35]. These similarities were expected as the studies were conducted a similar region (northwest Nigeria). However, as our study was conducted in a hospital setting with a large proportion of

**Table 6. Malnutrition estimates by simple/ acute necrotizing gingivitis diagnosis.**

| | | TOTAL | | Simple or acute necrotizing gingivitis | | No noma/ gingivitis | | Ch sq |
|---|---|---|---|---|---|---|---|---|
| **Chronic malnutrition (6 months-5 year olds)** | | N = 1909 | % | N = 52 | % | N = 1857 | % | <0.05 |
| | No | 954 | 50.0% | 19 | 36.5% | 935 | 50.4% | |
| | Yes | 955 | 50.0% | 33 | 63.5% | 922 | 49.7% | |
| | | | | | | | | |
| **Acute malnutrition (6 months-5 year olds)** | | N = 1910 | % | N = 49 | % | N = 1861 | % | 0.74 |
| | No | 548 | 28.7% | 13 | 26.5% | 535 | 28.8% | |
| | Yes | 1362 | 71.3% | 36 | 73.5% | 1326 | 71.3% | |
| **Acute malnutrition (6–12 year olds)** | | N = 98 | % | N = 6 | % | N = 92 | % | 0.59 |
| | No | 55 | 56.1% | 4 | 66.7% | 51 | 55.4% | |
| | Yes | 43 | 43.9% | 2 | 33.3% | 41 | 44.6% | |

**Table 7.** Risk factors for simple and acute necrotizing gingivitis on admission.

| | Total | | Simple gingivitis | | | | | Acute necrotizing gingivitis | |
|---|---|---|---|---|---|---|---|---|---|
| | N = 2346 | % | N = 58 | % | OR | 95% CI | P | N = 6 | % |
| **Patient age (years)** | | | | | | | <0.001 | | |
| 0 to 1 | 1097 | 47.1% | 12 | 20.7% | 1.00 | (Ref) | | 0 | 0.0% |
| 2 to 6 | 1135 | 48.8% | 41 | 70.7% | 3.38 | (1.77, 6.47) | | 4 | 66.7% |
| 7 to 12 | 95 | 4.1% | 5 | 8.6% | 5.03 | (1.73, 14.61) | | 2 | 33.3% |
| | 2327 | | 58 | | | | | 6 | |
| **Wealth score (patients family own: cell phone, radio, motorbike, tractor, bicycle, car)** | | | | | | | 0.728 | | |
| 0–2 items | 1696 | 72.3% | 43 | 74.1% | 1.00 | (Ref) | | 4 | 66.7% |
| 3–6 items | 650 | 27.7% | 15 | 25.9% | 0.90 | (0.50, 1.63) | | 2 | 33.3% |
| | 2346 | | 58 | | | | | 6 | |
| **ITFC -Child admitted** | | | | | | | 0.007 | | |
| No | 1269 | 54.1% | 21 | 36.2% | 1.00 | (Ref) | | 5 | 83.3% |
| Yes | 1077 | 45.9% | 37 | 63.8% | 2.10 | (1.22, 3.62) | | 1 | 16.7% |
| | 2346 | | 58 | | | | | 6 | |
| **Chronic malnutrition (6 months to 5 year olds)** | | | | | | | 0.148 | | |
| No | 958 | 50.1% | 19 | 39.6% | 1.00 | (Ref) | | 0 | 0.0% |
| Yes | 955 | 49.9% | 29 | 60.4% | 1.54 | (0.86, 2.77) | | 4 | 100.0% |
| | 1913 | | 48 | | | | | 4 | |
| **Acute malnutrition (6 months to 5 year olds)** | | | | | | | 0.761 | | |
| No | 550 | 28.7% | 12 | 26.7% | 1.00 | (Ref) | | 1 | 25.0% |
| Yes | 1364 | 71.3% | 33 | 73.3% | 1.11 | (0.57, 2.16) | | 3 | 75.0% |
| | 1914 | | 45 | | | | | 4 | |
| **Acute malnutrition (6–12 yr olds)** | | | | | | | 0.858 | | |
| No | 56 | 56.0% | 3 | 60.0% | 1.00 | (Ref) | | 1 | 100.0% |
| Yes | 44 | 44.0% | 2 | 40.0% | 0.85 | (0.13, 5.30) | | 0 | 0.0% |
| | 100 | | 5 | | | | | | |
| **This child has been sick in the past three months** | | | | | | | 0.942 | | |
| No | 1157 | 49.6% | 29 | 50.0% | 1.00 | (Ref) | | 1 | 16.7% |
| Yes | 1176 | 50.4% | 29 | 50.0% | 0.98 | (0.58, 1.65) | | 5 | 83.3% |
| | 2333 | | 58 | | | | | 6 | |
| **This child has had oral health issues in the past three months** | | | | | | | <0.001 | | |
| No | 2069 | 88.6% | 19 | 32.8% | 1.00 | (Ref) | | 2 | 33.3% |
| Yes | 265 | 11.4% | 39 | 67.2% | 18.75 | (10.65, 33.01) | | 4 | 66.7% |
| | 2334 | | 58 | | | | | 6 | |
| **The person answering the questions is the primary caretaker** | | | | | | | 0.817 | | |
| No | 266 | 11.4% | 6 | 10.3% | 1.00 | (Ref) | | 2 | 33.3% |
| Yes | 2070 | 88.6% | 52 | 89.7% | 1.11 | (0.47, 2.60) | | 4 | 66.7% |
| | 2336 | | 58 | | | | | 6 | |
| **Patient given colostrum at birth** | | | | | | | 0.201 | | |
| No | 669 | 28.7% | 21 | 36.2% | 1.00 | (Ref) | | 2 | 33.3% |
| Yes | 1663 | 71.3% | 37 | 63.8% | 0.70 | (0.41, 1.21) | | 4 | 66.7% |
| | 2332 | | 58 | | | | | 6 | |
| **Patient eats pap** | | | | | | | 0.163 | | |
| No | 589 | 25.2% | 10 | 17.2% | 1.00 | (Ref) | | 2 | 33.3% |
| Yes | 1746 | 74.8% | 48 | 82.8% | 1.63 | (0.82, 3.25) | | 4 | 66.7% |
| | 2335 | | 58 | | | | | 6 | |

*(Continued)*

**Table 7.** (Continued)

| | | Total | | Simple gingivitis | | | | | Acute necrotizing gingivitis | |
|---|---|---|---|---|---|---|---|---|---|---|
| | | N = 2346 | % | N = 58 | % | OR | 95% CI | P | N = 6 | % |
| **Patient vaccinated** | | | | | | | | 0.434 | | |
| | No | 922 | 39.5% | 20 | 34.5% | 1.00 | (Ref) | | 2 | 33.3% |
| | Yes | 1410 | 60.5% | 38 | 65.5% | 1.24 | (0.72, 2.15) | | 4 | 66.7% |
| | | 2332 | | 58 | | | | | 6 | |
| **Malaria- primary diagnosis** | | | | | | | | <0.001 | | |
| | No | 1246 | 53.5% | 46 | 79.3% | 1.00 | (Ref) | | 3 | 50.0% |
| | Yes | 1084 | 46.5% | 12 | 20.7% | 0.29 | (0.15, 0.55) | | 3 | 50.0% |
| | | 2330 | | 58 | | | | | 6 | |
| **Measles- primary diagnosis** | | | | | | | | 0.191 | | |
| | No | 2216 | 95.1% | 53 | 91.4% | 1.00 | (Ref) | | 5 | 83.3% |
| | Yes | 114 | 4.9% | 5 | 8.6% | 1.87 | (0.73, 4.77) | | 1 | 16.7% |
| | | 2330 | | 58 | | | | | 6 | |

OR = odds ratio; CI = confidence interval; P = p value from univariate logistic regression model

patients being treated for severe and moderate acute malnutrition with severe medical complications, it could have been expected that we would have seen more noma cases being diagnosed. One plausible explanation for this not being the case, is that the age range of our survey included a significant number of children aged under two years (47.1%), which is an age group less at risk of developing noma.

A study conducted in Cameroon in 2015 including school children (n = 2287) [38] found similar proportions of cases with gingivitis as in our study (2.1% had moderate gingivitis which could reflect our simple gingivitis cases and 0.6% had severe gingivitis which could reflect our acute necrotizing gingivitis cases), however comparison is difficult as the study methods and definitions differed significantly.

The parents/ guardians in our cohort reported several challenges accessing care; these are similar to challenges reported in other noma research work in this setting, including difficulties accessing transport and the long distances patients and their parents/ guardians need to travel in order to access care [39,40]. These issues should be taken into consideration for programs aiming to alleviate barriers to health care in this setting.

Many of the respondents who sought care for a prior illness first visited a street pharmacist (39.7%) showing that this group of health care workers are important agents for promoting oral health and awareness about noma. This finding is supported by another study in Northern Nigeria which showed that engaging community pharmacists may help to reduce oral health disparities by increasing oral health awareness and improving the quality of life via cost-effective delivery of pharmacy-based oral health-care services [41].

Many respondents reported cleaning their teeth (86.6%), however the main item used was water (84.7%), and only 10.4% reported using a toothbrush. The importance of tooth brushing and mechanisms to encourage and facilitate tooth brushing have been discussed in other literature [2,38,42–44]. Our findings showed that important aspects to include in oral health trainings in this setting are the benefits of cleaning, the importance of including toothbrushing in normal daily routines, and that toothbrushing will lead to having nice smelling breath and clean, bright teeth.

Respondents (n = 265, 11%) reported oral health issues in the past three months, and this was associated with simple and acute necrotizing gingivitis diagnosis. The oral health issues

reported (sore in the mouth, bleeding and sore gums) are all likely varying degrees of gingivitis and show that children living in the communities served by the hospital are experiencing poor oral hygiene. Our findings from the oral examinations on admission show lower rates of oral issues than those reported in the prior three months, which indicates that most oral complaints are transient, and not chronic or progressive.

Almost all the respondents (97.4%) had not had an oral health check in the past year. This low level of access and uptake of oral care is similar to other studies in the setting [45–48]. The main reasons for the lack of oral health care that have been reported are a lack of human resources, high costs of care, sparse provision of services and low levels of knowledge about different treatment options for oral conditions [45–48]. These barriers indicate the need to incorporate oral health into primary health care in this setting, a call common in oral health literature [46–48]. In terms of noma, it would be worthwhile encouraging the inclusion of oral examinations in primary health care provision, malnutrition surveys and vaccination campaigns, particularly in vulnerable populations.

Chronic malnutrition is associated with lower immunity, and those children admitted to the ITFC are malnourished and have a medical complication which likely means they are more immunocompromised than children not admitted to ITFC. This could explain the relationship between noma and being admitted to the ITFC. All of those diagnosed with acute necrotizing gingivitis had chronic malnutrition and most had acute malnutrition. This adds weight to the link between noma and malnutrition [8,10,11,29–31]. However, this link was not evident in our univariate regression analysis, one possible reason is that this analysis was conducted assessing risk factors for simple gingivitis, and most cases of simple gingivitis do not lead to later stage noma.

The most common primary diagnosis of patients admitted to the hospital were malaria (46.3%) and severe acute malnutrition (41.6%). Both of these conditions have been reported as risk factors for the development of noma [8,10,11,49]. However, our univariate analysis did not support this, possibly because our analysis was looking at risk factors for simple gingivitis, whereas the references are reporting risk factors for late stage noma. Another limitation is that our analysis was based on diagnosis at the same time as assessment of any stage of noma, whereas the references discuss diagnosis of comorbidities in the three months prior to onset of noma.

Our study was conducted in a hospital setting which in itself introduces certain limitations as the respondents only represent those children who were able to access care, which in this setting excludes those most at risk of developing noma (as a lack of access to care and low socioeconomic status are risk factors for noma). It is possible that children with the early stages of noma did not seek care at the hospital, which could lead to fewer cases of noma being identified in our study. These factors limit the generalisability of our findings and future studies should be conducted at a community level to better understand the epidemiology of noma.

Two factors influencing health care access during the study were the SARS-CoV-2 pandemic and the high levels of insecurity in the region, both could have changed the number and kinds of admissions at the hospital [50,51].

A further limitation was that we only recruited patients during the day time, this could have meant that patients with serious, acute complications (such as late stage noma) admitted at night time, were not included in our survey.

Our survey was conducted during the rainy season, which could have resulted in a possible bias, as this time of year could have meant increased difficulties for sicker patients to travel to the hospital which could have meant fewer later stage noma patients reached care. It could also have meant increased food insecurity, or difference in the epidemiology of other morbidities in the area. Future studies in other seasons or the year-round inclusion or oral screenings in

routine activities could offer insight into whether or not the season affects the number of simple and acute necrotizing gingivitis cases identified in this context.

## Conclusion

Our study showed a small proportion of those admitted to the Anka General Hospital, Zamfara, northwest Nigeria had simple or acute necrotizing gingivitis, most of which had resolved upon discharge. Those with chronic malnutrition and those admitted to the ITFC were more likely to have simple gingivitis, providing evidence of the link between malnutrition and the warning sign of noma. Many of those in our study cohort exhibited known risk factors for noma such as being malnourished, having a comorbidity, low socioeconomic status, poor oral hygiene practices, and a lack of access to healthcare, indicating there is a large population at risk of developing later stage noma in this setting. The lack of access to and uptake of oral health care indicates a strong need for oral exams to be included in routine health services. This provision could improve the oral status of the population and decrease the chance of patients developing later stage noma.

## Acknowledgments

Thank you to the participants in this study. A big thank you to the study team for all their hard work and dedication to this project. Thanks to Vincent Kimui Ndichu, Christian Mwemezi and Rahima Shuaibu for their contribution to the study. We would also like to thank all the health care workers who are advocating for and treating noma patients around the world.

## Author Contributions

**Conceptualization:** Elise Farley, Mark Sherlock, Joseph Samuel, Saskia van der Kam, Koert Ritmeijer, Cono Ariti, Mohana Amirtharajah, Annick Lenglet.

**Data curation:** Elise Farley, Miriam Njoki Karinja, Abdulhakeem Mohammed Lawal, Michael Olaleye.

**Formal analysis:** Elise Farley, Cono Ariti, Grégoire Falq.

**Funding acquisition:** Mark Sherlock, Deogracia Wa Kabila, Miriam Peters, Joseph Samuel, Guy Maloba, Mohana Amirtharajah, Annick Lenglet.

**Investigation:** Elise Farley, Miriam Njoki Karinja, Abdulhakeem Mohammed Lawal, Michael Olaleye, Sadiya Muhammad, Maryam Umar, Fatima Khalid Gaya, Shirley Chioma Mbaeri.

**Methodology:** Elise Farley, Cono Ariti, Mohana Amirtharajah, Annick Lenglet.

**Project administration:** Miriam Njoki Karinja, Abdulhakeem Mohammed Lawal, Michael Olaleye.

**Supervision:** Miriam Njoki Karinja, Abdulhakeem Mohammed Lawal, Michael Olaleye, Cono Ariti, Mohana Amirtharajah, Annick Lenglet, Grégoire Falq.

**Writing – original draft:** Elise Farley.

**Writing – review & editing:** Miriam Njoki Karinja, Abdulhakeem Mohammed Lawal, Michael Olaleye, Sadiya Muhammad, Maryam Umar, Fatima Khalid Gaya, Shirley Chioma Mbaeri, Mark Sherlock, Deogracia Wa Kabila, Miriam Peters, Joseph Samuel, Guy Maloba, Rabi Usman, Saskia van der Kam, Koert Ritmeijer, Cono Ariti, Mohana Amirtharajah, Annick Lenglet, Grégoire Falq.

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
