## [Decision Letter · Decision Letter 0]

25 Jul 2023

Dear Dr Farley,

Thank you very much for submitting your manuscript "Proportion of paediatric admissions with any stage of noma at the Anka General Hospital, northwest Nigeria" for consideration at PLOS Neglected Tropical Diseases. As with all papers reviewed by the journal, your manuscript was reviewed by members of the editorial board and by several independent reviewers. The reviewers appreciated the attention to an important topic. Based on the reviews, we are likely to accept this manuscript for publication, providing that you modify the manuscript according to the review recommendations. 

Sincerely,

Joseph M. Vinetz

Section Editor

Joseph Vinetz

Section Editor

Reviewer's Responses to Questions

**Key Review Criteria Required for Acceptance?**

**Methods**

-Are the objectives of the study clearly articulated with a clear testable hypothesis stated?

-Is the study design appropriate to address the stated objectives?

-Is the population clearly described and appropriate for the hypothesis being tested?

-Is the sample size sufficient to ensure adequate power to address the hypothesis being tested?

-Were correct statistical analysis used to support conclusions?

-Are there concerns about ethical or regulatory requirements being met?

Reviewer #1: The objectives of the study are clearly articulated. The study design has some limitations which I will describe in the general comment section, regarding the population and sample. No ethical concerns.

**Results**

-Does the analysis presented match the analysis plan?

-Are the results clearly and completely presented?

-Are the figures (Tables, Images) of sufficient quality for clarity?

Reviewer #1: The results are clear and figures/tables sufficient.

**Conclusions**

-Are the conclusions supported by the data presented?

-Are the limitations of analysis clearly described?

-Do the authors discuss how these data can be helpful to advance our understanding of the topic under study?

-Is public health relevance addressed?

Reviewer #1: The authors discuss some limitations. I recommend adding limitations in the general comments section. The public health relevance is clear.

**Editorial and Data Presentation Modifications?**

Reviewer #1: None

**Summary and General Comments**

Reviewer #1: Review PNTD-D-23-00848 PLOS Neglected Tropical Diseases 2023

Prospective study of children with any stage of noma admitted to a general hospital in northern Nigeria.

With the very limited research on noma, this prospective study is welcome and significant. Few other studies include the stage of noma, especially the early stages when the disease is easily treated, preventing both mortality and morbidity. This research aimed to investigate the role of malnutrition in the epidemiology of noma, with a view to advising prevention strategies. The research emphasizes the need for oral exams of young children and improving tooth cleaning based on good breath smell and white teeth, which appeals to the children and their caregivers. 

The abstract is concise. I recommend adding the dates and length of the research study.

Limitations were the study length for only five months in 2021, during the pandemic and the location in the hospital.

Did this hospital have any records of children with the diagnosis of acute noma or any evidence of noma survivors?

What about the limitations of the research located at the hospital?

Do children at this early stage visit the general hospital?

What about extremely poor children, without access to health care and malnourished?

Do you think this study was affected by the pandemic, limiting visits to the hospital ?

Do you have any recommendations to suggest for future studies to reach vulnerable children outside the hospital setting?

Noma is associated with children with chronic malnutrition, whereas this study targeted acute malnutrition. Did you investigate the proportion of children with chronic malnutrition (Height/age <-2SD, severe <-3SD)? Children with chronic malnutrition would not be admitted to the ITFC.

Line 144, page 8

I don't understand this sentence, please clarify. Does this mean that you know the proportion of pediatric admissions with noma? Is the word "than" correct?

Did you consider these two references?

Noma – a neglected disease of malnutrition

and poor oral hygiene: A mini-review

Wubishet Gezimu1 , Ababo Demeke2 and Abdissa Duguma1

SAGE Open Medicine

Volume 10: 1–5 2022

sagepub.com/journals-permissions

DOI: 10.1177/20503121221098110

Tickell KD, Walson JL (2016) Nutritional

Enteric Failure: Neglected Tropical Diseases and

Childhood Stunting. PLoS Negl Trop Dis 10(4):

e0004523. doi:10.1371/journal.pntd.0004523

Thank you for your dedication to address the gaps in knowledge of noma epidemiology. The aim to improve the nutritional and oral status of this vulnerable population and to recommend measures to prevent the development of noma are encouraged by your research and this publication. Thank you to PLOS NTDs for continuing to support noma research.

M. Leila Srour

MD FAAP MPH DTM&H

PLOS authors have the option to publish the peer review history of their article (what does this mean?). If published, this will include your full peer review and any attached files.

Reviewer #1: Yes: M. Leila Srour

Figure Files:

Data Requirements:

Reproducibility:

References

---

## [Editor Report · Decision Letter 1]

3 Oct 2023

Dear Dr Farley,

We are pleased to inform you that your manuscript 'Proportion of paediatric admissions with any stage of noma at the Anka General Hospital, northwest Nigeria' has been provisionally accepted for publication in PLOS Neglected Tropical Diseases.

Best regards,

Joseph M. Vinetz

Section Editor

Joseph Vinetz

Section Editor

---

## [Editor Report · Acceptance letter]

24 Oct 2023

Dear Dr Farley,

We are delighted to inform you that your manuscript, "Proportion of paediatric admissions with any stage of noma at the Anka General Hospital, northwest Nigeria," has been formally accepted for publication in PLOS Neglected Tropical Diseases.

Best regards,

Shaden Kamhawi

co-Editor-in-Chief

Paul Brindley

co-Editor-in-Chief
